

# Impact of Atmospheric and Aerosol Optical Depth Observations on Aerosol Initial Conditions in a strongly-coupled data assimilation system

Milija Zupanski[1], Anton Kliewer[1], Ting-Chi Wu[1], Karina Apodaca[1], Qijing Bian[2], Sam Atwood[2], Yi Wang[3,4,5], Jun Wang[3,4,5], Steven D. Miller[1]

[1]Cooperative Institute for Research in the Atmosphere, Colorado State University, Fort Collins, CO, USA

[2]Department of Atmospheric Science, Colorado State University, Fort Collins, Colorado USA

[3]Department of Chemical and Biochemical Engineering, The University of Iowa, Iowa City, IA USA

[4]Center of Global and Regional Environmental Research, The University of Iowa, Iowa City, IA USA

[5]Interdisciplinary Graduate Program in Informatics, The University of Iowa, Iowa City, IA USA

*Correspondence to*: Milija Zupanski (milija.zupanski@colostate.edu)

**Abstract.** Strongly coupled data assimilation frameworks provide a mechanism for including additional information about aerosols through the coupling between aerosol and atmospheric variables, effectively utilizing atmospheric observations to change the aerosol analysis. Here, we investigate the impact of these observations on aerosol using the Maximum Likelihood Ensemble Filter (MLEF) algorithm with Weather Research and Forecasting model coupled with Chemistry (WRF-Chem) which includes the Godard Chemistry Aerosol Radiation and Transport (GOCART) module.   We apply this methodology to

a dust storm event over the Arabian Peninsula and examine in detail the error covariance and in particular the impact of atmospheric observations on improving the aerosol initial conditions. The assimilated observations include conventional atmospheric observations and Aerosol Optical Depth (AOD) retrievals. Results indicate a positive impact of using strongly coupled data assimilation and atmospheric observations on the aerosol initial conditions, quantified using Degrees of Freedom for Signal.

## 1 Introduction

Efforts in assimilating atmospheric aerosol and chemistry observations into numerical models have involved mostly

variational data assimilation applications into global models (Zhang et al., 2008; Uno et al., 2008; Benedetti et al., 2009) and regional models (Pagowski et al., 2010; Liu et al., 2011). In addition, column-integrated aerosol data assimilation schemes (Weaver et al., 2007; Zhang et al., 2008) offer capabilities similar to the standard three-dimensional variational (3D-Var) and four- dimensional variational (4D-Var) methods, but with improved efficiency. One difficulty associated with variational methods is the modeling of forecast error covariance for aerosol/chemistry variables, prompting the use of Ensemble Kalman



Filter (EnKF) data assimilation methods (Pagowski and Grell, 2012; Rubin et al., 2016, 2017), which employ a flow-dependent error covariance. With the development of hybrid variational-ensemble methods, the improved representation of forecast error covariance and the ability of data assimilation to address nonlinear interactions have enabled the emergence of hybrid aerosol data assimilation applications (Schwartz et al., 2014; Pagowski et al., 2014).

Prediction models used for the simulation of aerosol fields are typically coupled with the atmosphere, and possibly chemistry, which creates an opportunity to employ strongly coupled data assimilation for advancing aerosol data assimilation in such complex systems (e.g., Zupanski, 2017). Un this research, we are mostly interested in aerosol data assimilation and prediction in the coastal regions. Since physical processes and interactions in the coastal zone are complex and time dependent, the representation of forecast error covariance in data assimilation should include a flow-dependent component. Therefore, an ensemble and/or hybrid ensemble-variational data assimilation system is preferable for such applications. In addition, a reliable three-dimensional picture of aerosol fields requires some information in the vertical. This can be achieved by using observations that have vertical distribution. However, Aerosol Optical Depth (AOD) observations assimilated in this work are vertically integrated and therefore do not include information about the vertical distribution of aerosol. A strongly coupled atmosphere-aerosol data assimilation frameworks provide a mechanism for including additional information in the vertical through the coupling between aerosols and atmospheric variables, effectively utilizing atmospheric observations to change the aerosol analysis. Since atmospheric observations are much more widely available than aerosol observations, and since they often include information in the vertical, in principle they can help improve 3-D aerosol analysis. As a side benefit, strongly coupled data assimilation (DA) acts as an additional constraint that controls adjustments in the analysis.

In this paper, we demonstrate the use of a strongly coupled atmosphere-aerosol data assimilation system and include a detailed examination of the coupled error covariance and the impact of atmospheric observations on improving aerosol initial conditions. The manuscript is organized as follows. In section 2 we present the data assimilation methodology and in section 3 we describe the coupled model and the observations. The experimental design is presented in section 4, results are discussed in section 5, and in section 6 we summarize and draw conclusions.

## 2 Data Assimilation Methodology

In this section we describe the mathematical basics of the employed data assimilation algorithm and of the strongly coupled DA.

### 2.1 Maximum Likelihood Ensemble Filter (MLEF)



The Maximum Likelihood Ensemble Filter (Zupanski, 2005; Zupanski et al., 2008) is a hybrid ensemble-variational method with explicit iterative minimization. It is well-suited for high-dimensional and nonlinear applications such as aerosol DA. It includes iterative minimization to deal with nonlinearities, similar to variational methods. However, unlike variational methods, it has an optimal Hessian preconditioning that not only provides very fast minimization (e.g., 1-2 iterations in most applications) but also a reliable estimate of the analysis error covariance used to define ensemble forecast initial perturbations. MLEF has been applied to high-resolution complex modelling systems that couple clouds, aerosols, carbon, and chemistry (Lokupitiya et al., 2008; S. Zhang et al., 2013; M. Zhang et al., 2013; Peters-Lidard et al., 2015; Park et al., 2015; Lim et al., 2015; Lee et al., 2017). Mathematical details of the MLEF algorithm can be found in Zupanski et al. (2008). Here we briefly mention relevant characteristics of MLEF.

The MLEF analysis is the maximum-a-posteriori (MAP) estimate obtained by minimizing the cost function

$$J(\mathbf{x}) = \frac{1}{2}[\mathbf{x} - \mathbf{x}^f]^T \mathbf{P}_f^{-1}[\mathbf{x} - \mathbf{x}^f] + \frac{1}{2}[\mathbf{y} - h(\mathbf{x})]^T \mathbf{R}^{-1}[\mathbf{y} - h(\mathbf{x})], \tag{1}$$

where $\mathbf{x}$ is a state vector, $\mathbf{y}$ is an observation vector, $h: S \rightarrow O$ is a nonlinear observation operator (where $\rightarrow$ denotes mapping, from space S to space O in this case), $S$ denotes the state space with dimension $N_s$, $O$ is the observation space with dimension $N_O$, $\mathbf{P}_f: S \rightarrow S$ is the forecast error covariance and $\mathbf{R}: O \rightarrow O$ is the observation error covariance. Superscript $T$ denotes the transpose, and $f$ denotes the forecast. Worth mentioning is that the first guess $\mathbf{x}^f$ in MLEF is obtained as a deterministic forecast from the previous analysis, not as an ensemble forecast mean used in ensemble Kalman Filtering methods. This might be advantageous for preserving a dynamical structure in high-resolution models, such as the coupled atmosphere-aerosol model used in this study.

Analysis uncertainty is estimated from the square root analysis error covariance

$$\mathbf{P}_a^{1/2} = \mathbf{P}_f^{1/2}(\mathbf{I} + \mathbf{Z}(\mathbf{x}^a)^T \mathbf{Z}(\mathbf{x}^a))^{-1/2}, \tag{2}$$

with the matrix $\mathbf{Z}$ defined as

$$\mathbf{Z}(\mathbf{x}^a) = [\mathbf{z}_1(\mathbf{x}^a), \quad \cdots, \quad \mathbf{z}_{N_E}(\mathbf{x}^a)], \tag{3}$$

$$\mathbf{z}_i(\mathbf{x}^a) = \mathbf{R}^{-\frac{1}{2}}[h(\mathbf{x}^a + \mathbf{p}_i) - h(\mathbf{x}^a)] \qquad (i = 1, \cdots, N_E), \tag{4}$$

where $\mathbf{p}_i$ are the columns of the square root forecast error covariance, $\mathbf{x}^a$ is the analysis, and $N_E$ is the ensemble space dimension. The formulation (2) also defines optimal Hessian preconditioning, using $\mathbf{x}^f$ instead of $\mathbf{x}^a$, however. After the





analysis is calculated by minimizing the cost function (1), the initial conditions for ensemble forecasting are obtained by adding the columns of the square root analysis forecast error covariance to the analysis

$$x_i = x^a + \left[P_a^{1/2}\right]_i \qquad (i = 1, \cdots, N_E). \tag{5}$$

After evolving the analysis $x^a$ and the perturbed initial conditions (5) using a prediction model $m$, the columns of the square root forecast error covariance in the next analysis time are

$$p_i = m\left(x^a + \left[P_a^{1/2}\right]_i\right) - m(x^a) \qquad (i = 1, \cdots, N_E). \tag{6}$$

In high-dimensional applications, such as the one conducted in this study, MLEF includes error covariance localization to increase degrees of freedom (e.g., Hamill and Whitaker, 2001). In particular, we use the method described by Yang et al. (2009).

**2.2 Strongly Coupled Data Assimilation**

15   Strongly coupled data assimilation means that forecast and analysis error covariances are completely defined from ensemble forecasting, implying that any correlations between the components (e.g., atmospheric and aerosol) will be kept. A standard analysis solution (e.g., Lorenc, 1986) can be written as

$$x^a = x^f + P_f H^T \left(H P_f H^T + R\right)^{-1} [y - h(x^f)] \tag{7}$$

where $H$ is the first derivative of a nonlinear observation operator $h$. The coupled forecast error covariance is a $2 \times 2$ block matrix

$$P_f = \begin{pmatrix} P_{atm,atm} & P_{atm,aero} \\ P_{atm,aero}^T & P_{aero,aero} \end{pmatrix}, \tag{8}$$

where indices *atm* and *aero* refer to the atmospheric and aerosol components, respectively. In strong coupling, the matrix of cross-component correlations $P_{atm,aero}$ can be non-zero. This means that the methodology allows non-zero correlations, but it does not necessarily imply that the actual correlations will be large, as they are defined by the dynamical system. For simplicity of presentation, we assume that aerosol and atmosphere observation operators are independent

30

$$H = \begin{pmatrix} H_{atm} & 0 \\ 0 & H_{aero} \end{pmatrix}. \tag{9}$$



Using (8) and (9), one can rewrite (7) in terms of its components

$$\begin{pmatrix} x_{atm}^a \\ x_{aero}^a \end{pmatrix} = \begin{pmatrix} x_{atm}^f \\ x_{aero}^f \end{pmatrix} + \begin{pmatrix} P_{atm,atm} & P_{atm,aero} \\ P_{atm,aero}^T & P_{aero,aero} \end{pmatrix} \begin{pmatrix} H_{atm}^T & 0 \\ 0 & H_{aero}^T \end{pmatrix} \begin{pmatrix} \widehat{y}_{atm} \\ \widehat{y}_{aero} \end{pmatrix}, \tag{10}$$

Where the $\widehat{y}$ vector represents scaled observations that already have some impact of cross-component correlations

$$\begin{pmatrix} \widehat{y}_{atm} \\ \widehat{y}_{aero} \end{pmatrix} = \left( H P_f H^T + R \right)^{-1} \begin{pmatrix} [y - h(x^f)]_{atm} \\ [y - h(x^f)]_{aero} \end{pmatrix}. \tag{11}$$

After further manipulation of (10) the component-wise analysis update is

$$x_{atm}^a = x_{atm}^f + P_{atm,atm} H_{atm}^T \widehat{y}_{atm} + P_{atm,aero} H_{aero}^T \widehat{y}_{aero} \tag{12}$$

$$x_{aero}^a = x_{aero}^f + P_{atm,aero}^T H_{atm}^T \widehat{y}_{atm} + P_{aero,aero} H_{aero}^T \widehat{y}_{aero}. \tag{13}$$

The above equations demonstrate the mechanism that allows atmospheric observations to impact the aerosol analysis, and vice versa. The relevant terms in this coupling are $P_{atm,aero} H_{aero}^T \widehat{y}_{aero}$ in (12) and $P_{atm,aero}^T H_{atm}^T \widehat{y}_{atm}$ in (13). The critical role of the cross-component covariance $P_{atm,aero}$ is evident: if it is equal to zero there is no coupling in the analysis, but if non-zero then observations of one component can impact the analysis of the other component.   Additional details
about strongly-coupled DA can be found in Zupanski (2017).

## 3 Model and Observations

### 3.1 WRF-Chem Model

The employed coupled atmosphere-aerosol model is the Weather Research and Forecasting model coupled with Chemistry (WRF-Chem) version 3.9.1.1. As a fully coupled regional meteorology–chemistry–aerosol model, WRF-Chem can simulate deposition, biogenic/anthropogenic emission, mixing, transport, chemical transformation, and aerosol interactions with
meteorological condition (Grell et al., 2005; Fast et al., 2006). Cloud microphysics is represented by the WRF-Single-Moment 6-class scheme (Hong and Lim, 2006) and convective and shallow clouds of sub-grid scale are parameterized using the Kain-Fritsch scheme (see Kain, 2004). The Community Atmospheric Model scheme (Collins et al., 2004) is also used,



which allows for the simulation of aerosols and trace gases and represents short- and long-wave radiative transfers in the atmosphere.

We chose the MOZART-4 gas-phase chemical scheme (Emmons et al. 2010) and GOCART bulk aerosol scheme (Ginoux et al., 2001; Chin et al., 2002) (WRF-Chem option *chem_opt*=12), referred to as MOZCART. The GOCART aerosol module simulates 14 aerosol species: hydrophobic and hydrophilic organic carbon and black carbon, sulphate, sea salt in four particle size bins with effective radii of 0.3, 1.0, 3.25, and 7.5 $\mu m$ for dry air, and dust particles in five particle size bins with effective radii of 0.5, 1.4, 2.4, 4.5, and 8.0 $\mu m$. The GOCART scheme calculates dust globally as a function of fraction of erodible area, porosity, and surface wind speed (Ginoux et al., 2001). Since MOZART-4 interacts with radiative transfer processes through calculation of photolysis that takes into account the impact of aerosols and clouds (Tie et al., 2003), GOCART is linked to radiative transfer processes indirectly through MOZART-4. Furthermore, GOCART itself has interaction with radiation related to photochemistry, and with cloud in terms of aqueous chemistry. The chemical initial and boundary conditions are obtained using the MOZBC utility which maps species concentrations from MOZART datasets to WRF-Chem concentrations.

The model computational domain is composed of a parent domain at 27 km resolution over $172 \times 138$ horizontal grids and a nested domain at 9 km resolution over $172 \times 169$ grid points, located over the Arabian Peninsula. Both domains include 50 vertical layers with a model top at 10 hPa.

**3.2 Atmospheric Observations**

The assimilated atmospheric observations are the conventional observations collected by the U.S. National Weather Service (NWS) for operational weather forecasting and assimilation. These observations are assimilated using the forward component of the Gridpoint Statistical Interpolation (GSI) system as an observation operator. The GSI observation operator is accessed from MLEF using a customized interface module. The benefit of using components of an operational weather system (e.g., GSI) is that quality control and observation errors are available and well-tested.

**3.3 Aerosol Observations**

The assimilated aerosol observations are AOD observations/retrievals obtained by combining the Moderate Resolution Imaging Spectroradiometer (MODIS) collection 6.1 (Sayer et al., 2014) with additional AOD observations over turbid coastal water regions developed by Wang et al. (2017). In particular we assimilate AOD at 0.55 μm wavelength with a resolution of 3-km. There are substantial data gaps during the daytime also, given the sampling of MODIS.




The AOD observation operator consists of two components: (i) interpolation from WRF-Chem grid points to AOD observation location, and (ii) transformation of the WRF-Chem environmental state (e.g., temperature, humidity, pressure, and GOCART aerosol species) to AOD. To compute total column integrated AOD within a model grid cell, we follow the equation given in Pagowski et al. (2014)

$$\tau(\lambda) = \sum_{i=1}^{n} \sum_{k=1}^{ktop} E_{ext} \times c_{ik} \times \rho_k \times d_k ,$$ (14)

where $c$ is aerosol mixing ratio, $\rho$ is the dry air density, $d$ is the layer depth, and $E_{ext}$ is the extinction coefficient. The extinction coefficient is pre-computed using Mie theory and accounts for hygroscopic growth (e.g., Petters and Kreidenweis, 2007). Index $i$ refers to aerosol species, while index $k$ denotes model layer.

## 4 Experimental Design

In this paper a dust storm event that occurred over the Arabian Peninsula on August 3-4, 2016 has been chosen. This storm is characterized by an interaction between two dust plumes that reside in two different air masses: one in a dry continental air mass over Saudi Arabian interior, and the other in a moist maritime air mass over the southeastern portion of the Arabian Peninsula. The development of the second plume produces a challenging air-sea interaction that has been addressed in other papers included in this special issue.

### 4.1 Experiments

Data assimilation experiments begin on August 3, 2016 at 0000 UTC and end on August 4, 2016 at 1800 UTC. The assimilation window is 6-hours, implying that there are 8 DA cycles. Atmospheric observations are available in every DA cycle, but AOD observations are available only during the daytime. This means that AOD observations were available only in DA cycles 2, 3, 6, and 7, which poses an additional challenge for data assimilation. Atmospheric observations are mostly surface measurements, with vertical information provided by some radiosonde observations. Since atmospheric and AOD observation operators are quasi-linear (only the vertical interpolation in the atmospheric observation operator is slightly nonlinear), the data assimilation performs only one minimization iteration.

The main data assimilation experiments include: (1) Assimilation of AOD observations only, and (2) Simultaneous assimilation of atmospheric and AOD observations. The second experiment is exploiting in full the strongly coupled DA capability for improving aerosol initial conditions, while the first experiment is a referent run. By comparing the two experiments we can assess the value of atmospheric observations for improving aerosol initial conditions in a strongly coupled data assimilation system. We also include a single AOD observation experiment to assess the structure of a coupled forecast error covariance.





The choice of control variables includes atmospheric and aerosol model variables. The atmospheric control variables are: air temperature, horizontal winds, surface pressure, and specific humidity, while the aerosol control variables include GOCART dust species only.

Observation errors used for AOD observations follow Pagowski (2014), distinguish between land and water, and make the error dependent on the value of observations. In particular, we use the following observation errors:

$$\sigma_{AOD} = \begin{cases} 0.1 + 0.15 \cdot y_{AOD} & (over\ land) \\ 0.1 + 0.05 \cdot y_{AOD} & (over\ water) \end{cases} \tag{15}$$

We also include AOD observation bias correction in data assimilation by adopting the moving-average approach (described by Kliewer et al. 2018).

**4.2 Verification**

Verification of data assimilation poses a challenge for the Arabian Peninsula domain, especially for aerosol. In general, it is preferred to use an independent data source for verification, i.e. data that are different from the assimilated data. Typical aerosol verifying observations include those from the Aerosol Robotic Network (AERONET) (https://aeronet.gsfc.nasa.gov) and CALIOP data (e.g., Toth et al. 2013) onboard the CALIPSO satellite. Unfortunately, only 1-2 AERONET sites were available during this period, and the CALIPSO satellite has an infrequent passage over the area of interest. Data assimilation is designed to minimize the cost function (1) over the whole analysis domain, consequently verifying its impact at only few points is not adequate. Therefore, we opt for other means of verification. We include estimates of Degrees of Freedom for Signal (DFS) to quantify the impact of data assimilation (e.g., Zupanski et al. 2007). In most general form, DFS refers to reduction of Shannon entropy due to data assimilation (e.g., Rodgers 2000). The MLEF algorithm produces DFS as a by-product, and these are represented by

$$DFS = \sum_i \frac{\lambda_i}{1+\lambda_i} \tag{16}$$

where $\lambda$ are the eigenvalues of the matrix $Z = \left(R^{-1/2}HP_f^{1/2}\right)^T\left(R^{-1/2}HP_f^{1/2}\right)$. Since the matrix Z is real and positive-semidefinite definite, the DFS values are non-negative: zero values indicate null impact of data assimilation, while positive values quantify the efficiency of data assimilation in using information from the assimilated observations. A theoretical maximum calculated assuming completely independent ensemble perturbations and sufficient number of observations is approximately equal to one half of the ensemble size. In our experiments with 32 ensembles this would be around 16.





However, ensemble perturbations are not fully independent and there are grid points with small number of neighbouring observations, therefore further reducing the achievable DFS maximum value.

## 5 Results

All results presented here are for the inner domain of the WRF-Chem model, at a horizontal resolution of 9 km.

### 5.1 Observations

As mentioned earlier AOD observations had both temporal and spatial gaps. In Fig. 1 we show the AOD observation coverage when they were available. One can notice generally satisfactory coverage over the Persian Gulf, with the exception of August 4, 2016 at 0600 UTC due to the sun glint, since no retrievals are conducted over water in the glint zone.

In Table 1 we include the number of atmospheric and AOD observations that passed basic quality control. One can see that the number of atmospheric observations is largest at 0000 UTC, while the number of AOD observations varies. Most noticeable is a drop in the number of AOD observations on August 4, 2016 at 0600 UTC, due to lack of nighttime retrievals. Atmospheric observations, although small in number, are available at each analysis time. This underlines the need for a mechanism that can transfer the information from atmospheric observations to aerosol variables, which in this paper is addressed through the strongly coupled data assimilation framework.

### 5.2 Assimilation of a Single AOD Observation

Forecast error covariance has a fundamental role in any data assimilation methodology. In strongly coupled data assimilation its role is even more critical, as cross-component covariance provides a mechanism for improving the benefit of observations, as explained in section 2.2. Assessing its structure in a given application is a good indicator of the potential impact of strongly coupled data assimilation. This is typically achieved by conducting a data assimilation experiment with the assimilation of a single observation (e.g., Thepaut et al. 1996; Park et al. 2015). Therefore, we conduct an assimilation experiment in which a single AOD observation is assimilated. All specifications of the model and data assimilation remain as described in section 3.2. The single AOD observation is located at $27.5° N$ and $50.5° E$. We choose the DA cycle 7, valid on August 4, 2016 at 1200 UTC, so that the coupled forecast error covariance had time to develop its dynamical structure. In Fig.2 we show the analysis increments $(x^a - x^f)$ for the DUST-1 GOCART variable (effective radii of $0.5$ $\mu m$). One can notice the well-defined and localized impact of the AOD observation on dust. Fig.2 closely describes the structure of the matrix $P_{aero,aero}$ from section 2.2.



We are particularly interested in the cross-component covariance matrix $P_{atm,aero}$. The structure of this matrix is shown in Fig.3, for temperature, specific humidity, and horizontal wind, all at model level 8. The plots once again show well-defined and localized structures, indicating that cross-component covariance has a favourable structure for improving the utility of AOD observations. Temperature clearly shows a negative increment, while specific humidity shows a positive increment. A dipole structure can be seen in the wind increment, suggesting an increased convergence at observation location.

An interpretation of the impact of a single AOD observation in data assimilation can be as follows: a higher value of an AOD observation implies an increase of dust throughout the lower troposphere, which has a cooling effect that is also facilitated by temperature advection away from the observation location. Reduction of temperature implies reduced atmospheric pressure and eventually higher specific humidity. Although other explanations may be possible, this interpretation suggests that data assimilation has a dynamic response to an observation, a consequence of ensemble-based coupled forecast error covariance. The above discussion also indicates that structure of the cross-component covariance is satisfactory and possibly beneficial for strongly coupled data assimilation.

## 5.3 Assimilation of All Available Data

### 5.3.1 Degrees of Freedom for Signal

We now present the results of strongly coupled data assimilation that considers all available data. We first focus on the DFS measure to check the performance of coupled data assimilation in terms of entropy reduction. In order to facilitate the comparison with the reference run (i.e. assimilation of AOD observations only) we show DFS at DA cycles with AOD observations in Fig.4. The granulated structure of DFS is a consequence of covariance localization, which implies that DFS is calculated only at coarser analysis points. Although DFS could be interpolated to fine resolution points resulting in a smoother DFS field, we are keeping a constant DFS value for all fine resolution points surrounding the actual analysis point so that the coarse resolution of analysis points can be seen. There are several things that can be noticed in Fig.4, such as that higher values of DFS coincide with the location of observations. This also implies that there is a reduced DFS value over the Persian Gulf on August 4 at 0600 UTC, caused by the earlier mentioned nocturnal gap in AOD observations. One can also notice a steady increase of DFS value over time. Considering a higher number of total observations on August 3 at 0600 UTC and 1200 UTC, one would think that higher DFS should be obtained at those analysis times. However, Fig.4 shows a relatively larger DFS on August 4 at 0600 UTC and 1200 UTC. This essentially states that the data assimilation algorithm was able to extract more information from observations at later cycles. A possible interpretation of these results is that strongly coupled error covariance was more realistic at later DA cycles, making the impact of atmospheric, as well as AOD observations, more efficient.





Although the DFS results in strongly coupled assimilation of atmospheric and AOD observations are indicating an improvement due to assimilation, it is still not clear if atmospheric observations were helpful. The following is a central question of the strongly coupled data assimilation system: is the ensemble-based strongly coupled forecast error covariance sufficiently realistic in representing the cross-component correlations in our experiments? Results from our single AOD assimilation experiments suggested this is the case, but only the actual assimilation of all observations can confirm it. Therefore, we compare the DFS results from Fig.4 to DFS obtained in the assimilation of AOD observations only experiment by creating a difference $DFS_{ATM+AOD} - DFS_{AOD}$, shown in Fig.5. Positive values of this difference indicate positive impact of atmospheric observations, while negative values indicate a degradation of the analysis due to the impact of strongly coupled error covariance. One can see both the positive and negative values. However, the covered area and the magnitudes of positive values are clearly dominant, indicating that atmospheric observations had an overall positive impact in data assimilation. It is especially important to note the strong positive impact in the northern part of the Persian Gulf on August 4 at 0600 UTC. This is the analysis time when there was a gap in AOD observations, practically over the entire gulf (e.g., Fig.1). At this time the AOD only assimilation could not produce any impact in this area simply because there were no observations in the vicinity of that area. On the other hand, assimilation of combined atmospheric and AOD observations did produce a positive impact, implicitly suggesting that the strongly coupled ensemble-based forecast error covariance was able to transfer the information from atmospheric observations.

Both experiments, with and without the assimilation of atmospheric observations, show non-zero values of DFS. Note that in an experiment without assimilation of any data DFS would be equal to zero. Therefore, positive values of DFS confirm that data assimilation had a positive impact in either experimental configuration.

### 5.3.2 Aerosol Analysis Increments

After showing the positive impact of strongly coupled data assimilation in terms of DFS, it is important to understand where the dominant changes occurred, especially in the vertical. Note that even a stand-alone aerosol data assimilation produces a three-dimensional adjustment of aerosols. As shown in Fig.6, the reference experiment with assimilation of only AOD observations produces DUST-1 analysis adjustments throughout the lower troposphere. In the analysis field one can notice a mid-tropospheric layer with lofted dust up to 600 hPa, reaching the maximum at about 800 hPa. The cross-section of DUST-1 analysis increment (analysis-minus-guess) shows also a vertical structure with noticeable adjustments up to 500 hPa. This illustrates the ability of data assimilation to adjust vertical aerosol fields even though the assimilated observations only have a horizontal structure. Without any other influences, this vertical adjustment in data assimilation happens in agreement with the vertical distribution of aerosol uncertainty in the forecast guess.

Since our special interest is in the impact of atmospheric observations on aerosol initial conditions in strongly coupled atmosphere-aerosol data assimilation, we are interested in knowing whether or not atmospheric observations can impact the



adjustment of aerosols, especially in the vertical. Therefore, in Fig.7 we show a difference between DUST-1 analysis in the DA experiment that includes the assimilation of atmospheric and AOD observations and the experiment with assimilation of AOD observations only. The difference represents an accumulated impact of atmospheric observations. One can see that there is a noticeable impact of atmospheric observations on the aerosol vertical distribution, with maximum adjustments
reaching approximately ±20 μg/kg-dryair. From the horizontal map in Fig.7 there is a well-defined positive impact over the Persian Gulf, with positive DUST-1 increments outlining the strengthening of the dust plume due to assimilation of atmospheric observations. The vertical cross-section in Fig.7 suggests that atmospheric observations had the most pronounced impact in the 900-600 hPa layer, mostly strengthening the dust lofting.

### 5.3.3 Aerosol Uncertainty

By its construct data assimilation will produce a reduction of uncertainty in any given DA cycle. However, unrealistic uncertainty reduction would prevent development of a reliable forecast error covariance in continuous data assimilation
cycles. The single observation experiments and the DFS results suggest that covariance improvement was realistic, possibly implying that uncertainty reduction was also realistic. In Fig.8 we show the uncertainty of DUST-1 in terms of the standard deviation of the forecast error from the reference experiment, with assimilation of AOD observations only, and from the strongly coupled DA experiment, with the assimilation of atmospheric and AOD observations. One can see that strongly coupled DA experiment results in additionally reduced aerosol uncertainty compared to the referent DA experiment.
Although uncertainty reduction is generally seen throughout the domain, lower values over the north-western part of the Persian Gulf, where the storm was located at 1200 UTC on August 4, 2016, are especially relevant. There is about 25% reduction of dust uncertainty in this region due to the assimilation of atmospheric observations. Overall, Fig.8 suggests that atmospheric observations had a positive impact in reducing the uncertainty of dust estimates.

### 6 Summary and Conclusions

In this paper we presented preliminary results of strongly coupled atmosphere-aerosol data assimilation with the WRF-Chem modelling system that includes the GOCART aerosol module, in a dust storm application over the Arabian Peninsula. The
assimilated observations include conventional atmospheric observations and AOD retrievals.

Of special interest in this study was the use of strongly coupled data assimilation system capable of having atmospheric observations impacting the aerosol. Preliminary investigation of strongly coupled error covariance in this system was found to be satisfactory, with generally well-defined cross-component structure. The performance of this system in terms of DFS
indicative of the positive impact of data assimilation in all experiments. Another main goal of this study was to assess the



impact of atmospheric observations on aerosol initial conditions in a strongly coupled data assimilation system. It was found that the impact of atmospheric observations on aerosols is generally positive. In terms of DFS there is an improved efficiency of data processing, resulting in a higher DFS values when atmospheric data are assimilated. In terms of the three-dimensional structure of the aerosol field, it was found that atmospheric observations have additional impact on the vertical

distribution of the dust analysis adjustment. A reduction of dust uncertainty could also be seen when atmospheric observations are assimilated.

The use of a strongly coupled data assimilation system requires additional assessment and possible improvement. For example, strongly coupled error covariance is currently controlled only by covariance localization only. This can be likely

improved if the cross-component correlations between relevant variables is kept, while reducing or setting to zero the weak correlations between other variables. We plan to investigate in detail this possibility in the near future.

We are also interested in applying the methodology presented here to coastal areas at higher resolutions of 1-3 km. Assimilation of additional aerosol observations is also planned, with night time AOD data and other datasets that provide a

vertical distribution of aerosols.

*Author contributions.* MZ and AK contributed in developing the aerosol observation operator with AK developing the associated look-up table with help from TCW. KA helped in preliminary design of the experiments as well as in providing the expertise on

aerosol-atmosphere interactions. QB and SA contributed in determining $E_{ext}$ for the GOCART aerosol species after experiencing hygroscopic growth. YW and JW provided AOD observations used in data assimilation. SM provided the Mie-theory code used in building the look-up table. Majority of experiments was conducted and written by MZ, with consultation from all other authors.

*Competing interests.* The authors declare that they have no conflict of interest.

*Acknowledgements.* This work is supported by the Office of Naval Research (ONR) Multidisplinary University Research Initiative (MURI) program grant N00014-16-1-2040.

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





**Table 1**: Number of observations available for data assimilation.

| Date | Number of Atmospheric Observations | Number of AOD Observations |
|---|---|---|
| August 3, 2016 at 0000 UTC | 704 | - |
| August 3, 2016 at 0600 UTC | 286 | 3149 |
| August 3, 2016 at 1200 UTC | 438 | 3297 |
| August 3, 2016 at 1800 UTC | 240 | - |
| August 4, 2016 at 0000 UTC | 896 | - |
| August 4, 2016 at 0600 UTC | 261 | 797 |
| August 4, 2016 at 1200 UTC | 382 | 1248 |
| August 4, 2016 at 1800 UTC | 224 | - |





**Figure 1: AOD observation coverage during data assimilation. Observations are available only at daylight, at 0600 UTC and 1200**
30 **UTC. DA cycles at night time correspond to 1800 UTC and 0000 UTC, when AOD observations were not available.**





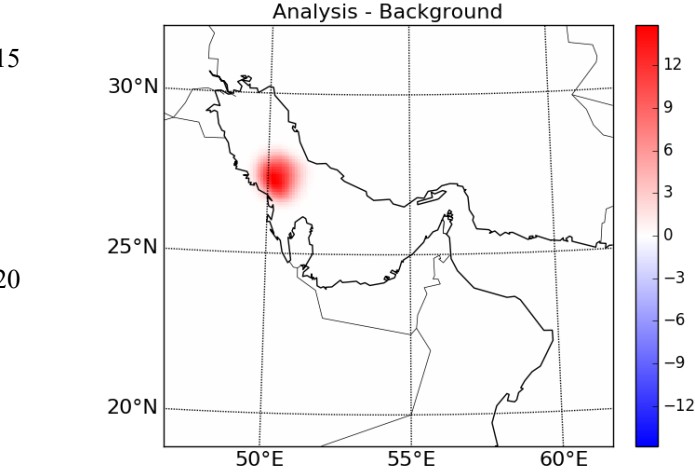 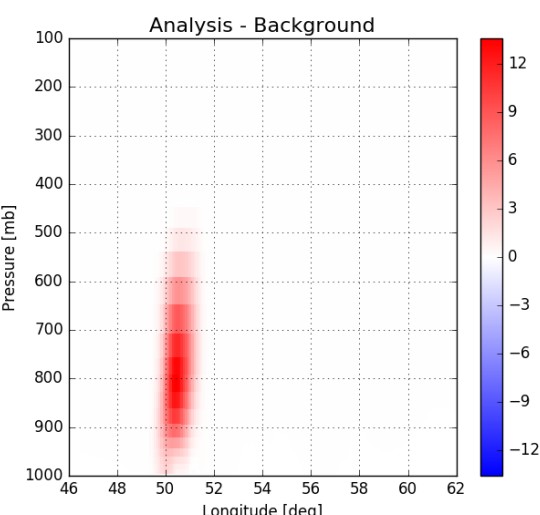

**Figure 2: Analysis increments of DUST-1 GOCART variable (μg/kg-dry air) in single AOD observation experiment valid August 4, 2016 at 1200 UTC: horizontal map at model level 5, near surface (left panel), and vertical cross-section at 27.5° *N* (right panel).**

30

35





**Figure 3: Analysis increments at model level 8 in single AOD observation experiment valid August 4, 2016 at 1200 UTC: temperature (K) (upper left panel), specific humidity (kg/kg) (upper right panel), and horizontal wind (m/s) (bottom panel).**

30





**Figure 4: DFS in strongly coupled data assimilation of atmospheric and AOD observations at model level 5.**





**Figure 5: DFS difference between the experiment with assimilation of all (e.g., atmospheric and AOD) observations and with assimilation of AOD observations only.**



**Figure 6: Vertical cross-section of DUST-1 (µg/kg-dryair) at 27.5° _N_ in the experiment with assimilation of AOD observations only: analysis (left panel), and analysis increment (analysis-minus-guess) (right panel), valid 1200 UTC on August 4, 2016.**

30

35



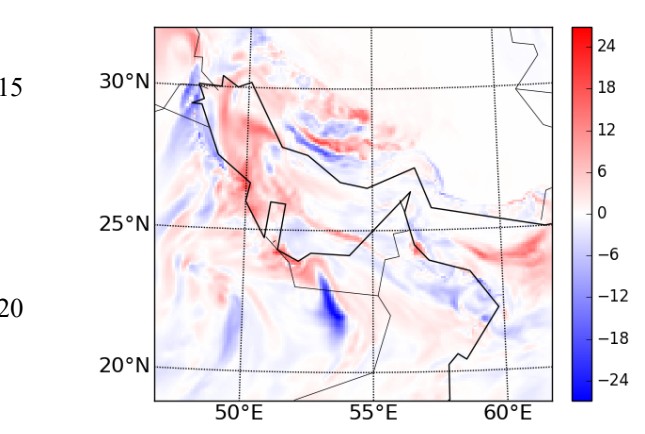 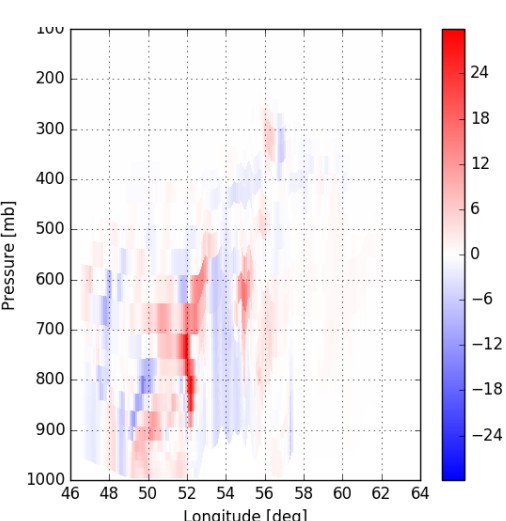

**Figure 7: DUST-1 difference (µg/kg-dryair) between the experiments with assimilation of atmospheric _and_ AOD observations, and assimilation of AOD observations only, valid 1200 UTC on August 4, 2016: horizontal difference at Lev=5 (left panel) and difference in vertical at $27.5°\,N$ (right panel).**

30

35



**Figure 8: Forecast error standard deviation of DUST-1 (µg/kg-dryair) at Lev=5, valid 1200 UTC on August 4, 2016: assimilation of**
25 **AOD observations only (left panel) and assimilation of atmospheric and AOD observations (right panel).**