# Peer review of "Impact of Atmospheric and Aerosol Optical Depth Observations on Aerosol Initial Conditions in a strongly-coupled data assimilation system"

_Atmospheric Chemistry and Physics, 2019_

## Referee Comment (RC1) · Anonymous Referee #1 · 25 Apr 2019

The paper presents a demonstration of a strongly coupled assimilation system for the atmosphere with aerosols. It uses two established systems, for the atmospheric modeling using WRF-Chem, and for assimilation the hybrid-ensemble MLEF system. The main weaknesses of the paper are brought up by the authors themselves, in general independent data is used for verification, and though the DFS is informative and can show general indications of system performance a broader case study and exercise is needed with traditional verification using either an independent model or observations. The interesting case presented for the dust plume on the 4th of August should be included; however, this should be just one of many or part of a much longer examination of the system performance. In the conclusions, the authors state that additional

assessment is required, and there is little presented on the meteorological variables which may or may not have been impacted by the AOD assimilation. Again thorough examination of these meteorological variables against either another modeling system or independent data is needed to complete the analysis of the system performance. Further, there are 14 species in GOCART, and only DUST-1 has been presented, it would be enlightening to know the impacts on these different species as well. Regrettably there is much needed yet in this manuscript to bring it up to a full examination of the system and its performance. At this time I recommend rejection of the current manuscript and encourage a resubmission when the WRF-Chem/MLEF system has been explored and exercised in more depth.

Small typographical corrections: page 2, line 8. "In this research" page 12, line 19 ". . . experiment results additionally reduced" page 7, line 33 and page 12 line 18 "referent" experiment was used while other times "reference" you may consider the use of "reference" consistently. Comment on 5.3.3 Aerosol Uncertainty and associated Figure 8: regarding the lowering of the forecast error standard deviation for DUST-1, couldn't this just be due to the inclusion of the atmospheric observation making the ensemble members more consistent with one another with the wind forcing and placement of the dust. It seems obvious that this may occur, but again the exploration of any unexpected correlations is what needs to be explored.

---

## Referee Comment (RC2) · Anonymous Referee #2 · 29 Apr 2019

This article is ostensibly about the impact of using a coupled aerosol-atmospheric model in data assimilation. Unfortunately, one can not measure the impact of a change on a data assimilation system without proper verification. The "Degrees of Freedom for Signal", which is used as the dominant metric for success in this article, is not a sensible measure to use for verification as it does not measure ones distance from truth (or, in practice, observations). I suppose one could use this measure in a secondary role if one had already proven through traditional verification that the system was performing very well. The authors attempt to side step this issue on page 8 by stating that verification poses a "challenge", and while I agree it does in this case, it nevertheless still stands that we can't know the impact without verification against observations.

Similarly, showing analysis increments and measures of ensemble derived uncertainty misses the mark as these two measures as well as the DFS all can be manipulated to appear successful even when a data assimilation system is doing poorly. For these reasons I must recommend rejection of this article, but also I would encourage the author's to spend the time to construct proper verification against observations.

---

## Author Comment (AC1) · 11 Jul 2019

**Authors' response to Referees**

We thank both Referees for constructive and helpful comments. We responded to all comments, closely following Referees' suggestions.

The main issue brought up by Referees is the lack of verification against independent observations. We answer to that below but would like to make a general comment that this issue will frequently appear in applications with new type of measurements. Instead of avoiding such applications because of the lack of independent data, we believe one should explore such novel applications and in process try to develop alternative verification measures.

Since employing standard methods of verification using independent observations in data scarce areas is simply not possible, alternative methods for verification and comparison of results are needed. We believe that DFS is one of those measures. Its relationship to errors with respect to independent data is now described in new Appendix. It is shown in simple mathematical terms that DFS and errors with respect to independent observations cannot go in different directions, as they are tied to each other by the influence matrix $\boldsymbol{HK}$. From the Appendix, the analysis error is

$$\boldsymbol{\varepsilon}_a = \boldsymbol{HK}\boldsymbol{\varepsilon}_o + \left(\boldsymbol{I}_p - \boldsymbol{HK}\right)\boldsymbol{\varepsilon}_b$$

and DFS is

$$d_s = tr(\boldsymbol{HK})$$

A possible reason why DFS is sometimes considered as "unreliable" and "not sensible" is because it is difficult to compute in most DA algorithms and little exploration of its value has been done. However, in MLEF, as well as in ETKF and LETKF data assimilation algorithms, DFS is a by-product of eigenvalue decomposition used in matrix inversion, and therefore readily available.

Therefore, we strongly believe that fundamental role of the influence matrix in data assimilation supports the use of DFS in verification. We do agree that more research related to the value of DFS and its relationship with analysis error is needed, and this would be one of the future topics we plan to undertake, thanks to the Referees' comments.

*Referee-1*

*The paper presents a demonstration of a strongly coupled assimilation system for the atmosphere with aerosols. It uses two established systems, for the atmospheric modeling using WRF-Chem, and for assimilation the hybrid-ensemble MLEF system. The main weaknesses of the paper are brought up by the*

*authors themselves, in general independent data is used for verification, and though the DFS is informative and can show general indications of system performance a broader case study and exercise is needed with traditional verification using either an independent model or observations.*

As we mention above, employing standard methods of verification using independent observations in data scarce areas is simply not possible. The approach taken here was to rely on DFS instead, which we believe is equally important as errors with respect to independent observations. The supporting mathematical arguments are presented in new Appendix.  In addition to DFS, we include verifications of 6-hour forecasts from data assimilation against ICAP MME reanalysis and NUCAPS retrievals. We also calculate RMS errors with respect to observations used in assimilation (Table 2).

The verification of data assimilation is further clarified in section 4.2.

New verifications are added, for both the aerosol (Figs. 6, 7, Table 2) and the meteorological fields (Fig. 8), that include ICAP MME reanalysis, RMS errors with respect to observations, and NUCAPS retrievals.

We also include a new paragraph (p. 3, lines 10 – 23) describing the previous successful use of MLEF algorithm in similar applications.

*The interesting case presented for the dust plume on the 4th of August should be included; however, this should be just one of many or part of a much longer exam nation of the system performance.*

This manuscript is part of the Special Issue, and other manuscripts give a more detailed description of the dust plumes and synoptic details. Following Referee's suggestion, we include a more detailed description of the dust plume case by introducing a new section (4.1) and new plots (Figs. 1, 2).

*In the conclusions, the authors state that additional assessment is required, and there is little presented on the meteorological variables which may or may not have been impacted by the AOD assimilation. Again thorough examination of these meteorological variables against either another modeling system or independent data is needed to complete the analysis of the system performance.*

Verification of meteorological variables using NUCAPS retrievals is now included (p. 13, last paragraph), with new Fig. 8. We note, however, that this manuscript focuses on the impact of meteorological observations on aerosol, not on meteorology itself. This is relevant in the sense of error covariance, as we are interested in the value of the cross-component correlations, potentially difficult to accurately represent, rather than the auto-correlations which are generally easier to get.

*Further, there are 14 species in GOCART, and only DUST-1 has been presented, it would be enlightening to know the impacts on these different species as well. Regrettably there is much needed yet in this manuscript to bring it up to a full examination of the system and its performance. At this time I recommend rejection of the current manuscript and encourage a resubmission when the WRF-Chem/MLEF system has been explored and exercised in more depth.*

Given the specifics of the geographical area and the chosen model domain, only dust is included as the aerosol control variable. Our earlier assessment indicated that other aerosol species are essentially negligible in this case, prompting the choice of dust only. Following Referee's suggestions we added results and discussion for DUST-5 (p. 15, lines 20-24, p.16, lines 1-5, Figs. 11c,d and 12c,d).

*Small typographical corrections: page 2, line 8. "In this research" page 12, line 19 ". . . experiment results additionally reduced" page 7, line 33 and page 12 line 18 "referent" experiment was used while other times "reference" you may consider the use of "reference" consistently.*

We changed the typos noticed by the Referee.

*Comment on 5.3.3 Aerosol Uncertainty and associated Figure 8: regarding the lowering of the forecast error standard deviation for DUST-1, couldn't this just be due to the inclusion of the atmospheric observation making the ensemble members more consistent with one another with the wind forcing and placement of the dust. It seems obvious that this may occur, but again the exploration of any unexpected correlations is what needs to be explored.*

We agree with the Referee that uncertainty analysis was inconclusive, and we decided to exclude it from the manuscript, as well as the original Figure 8. We also felt that shortening the manuscript by excluding this topic would somewhat balance the increased manuscript size with all new figures and text related to verification.

**Referee-2**

*This article is ostensibly about the impact of using a coupled aerosol-atmospheric model in data assimilation. Unfortunately, one can not measure the impact of a change on a data assimilation system without proper verification. The "Degrees of Freedom for Signal", which is used as the dominant metric for success in this article, is not a sensible measure to use for verification as it does not measure ones distance from truth (or, in practice, observations). I suppose one could use this measure in a secondary role if one had already proven through traditional verification that the system was performing very well. The*

***authors attempt to side step this issue on page 8 by stating that verification poses a "challenge", and while I agree it does in this case, it nevertheless still stands that we can't know the impact without verification against observations.***

As we mention above, employing standard methods of verification using independent observations in data scarce areas is simply not possible. The approach taken here was to rely on DFS instead, which we believe is equally important as errors with respect to independent observations. The supporting mathematical arguments are presented in new Appendix.

When the Referee mentions a "distance from truth", and lack of it from DFS, we would like to add that distance is generally defined in terms of a norm. This can be not only the frequently used Euclidian norm, but also the trace norm (e.g., Shatten p-1 norm) used in DFS definition.

In addition to DFS, we include verifications of 6-hour forecasts from data assimilation against ICAP MME reanalysis and NUCAPS retrievals. We also calculate RMS errors with respect to observations used in assimilation (Table 2).

The verification of data assimilation is further clarified in section 4.2.

New verifications are added, for both the aerosol (Figs. 6, 7, Table 2) and the meteorological fields (Fig. 8), that include ICAP MME reanalysis, RMS errors with respect to observations, and NUCAPS retrievals.

We also include a new paragraph (p. 3, lines 10 – 23) describing the previous successful use of MLEF algorithm in similar applications.

***Similarly, showing analysis increments and measures of ensemble derived uncertainty misses the mark as these two measures as well as the DFS all can be manipulated to appear successful even when a data assimilation system is doing poorly. For these reasons I must recommend rejection of this article, but also I would encourage the author's to spend the time to construct proper verification against observations.***

We do not agree with the Referee that DFS can be "manipulated" as it is tied to analysis error by the influence matrix. This is also supported by new verifications, for both the aerosol (Figs. 6, 7, Table 2) and the meteorological fields (Fig. 8), that include ICAP MME reanalysis, RMS errors with respect to observations, and NUCAPS retrievals.

Prompted by the Referee's comments and the encouragement to further explore alternative verifications, we are now developing a formal mathematical foundation for verification without independent observations that shows that DFS and analysis errors have very tight bounds that fully control each other's behavior.

**Authors' changes in manuscript**

This is the list of changes to the original manuscript:

1) p. 3, lines 10-22: previous experience with MLEF applied to similar problems.

2) p.8, section 4.1, Figs. 1 and 2: New sub-section added to describe dust storm. Two figures (Figs. 1 and 2) are also added.

3) p.8-11, section 4.2: A more detailed description of verification measures is included, with new Appendix.

4) p.12-14, section 5.3.1: New sub-section is added related to verification of data assimilation. This also includes new Figures 6, 7, and 8, and new Table 2.

5) p.15, lines 20-23: Additional detail related to DUST-5, with new Figure 11c,d.

6) p.16, lines 1-5: Additional detail related to DUST-5, with new Figure 12c,d.

7) Deleted the original sub-section 5.3.3 and Figure 8, related to aerosol uncertainty discussion.

8) New Appendix (p.17-19) is added to address the mathematical relationship between DFS and errors with respect to independent observations.

9) Additional references (Cardinali et al. 2004, Desroziers et al. 2010, Miller et al. 2019, Talagrand 1997, Sessions et al. 2015).